# Neuropsychological Characteristics and Quantitative Electroencephalography in Skogholt’s Disease—A Rare Neurodegenerative Disease in a Norwegian Family

**DOI:** 10.3390/brainsci14090905

**Published:** 2024-09-06

**Authors:** Knut A. Hestad, Jan O. Aaseth, Juri D. Kropotov

**Affiliations:** 1Department of Research, Innlandet Hospital Trust, N-1381 Brumunddal, Norway; jaol-aas@online.no; 2N.P. Bechtereva Institute of the Human Brain, Russian Academy of Sciences, 194064 St. Petersburg, Russia; yurykropotov@yahoo.com

**Keywords:** Skogholt’s disease, neuropsychology, QEEG, ERP, neurodegenerative

## Abstract

Members of three generations of a Norwegian family (N = 9) with a rare demyelinating disease were studied. Neuropsychological testing was performed using the Mini Mental Status Examination (MMSE), Wechsler Intelligence Scale-III (WAIS-III), and Hopkins Verbal Learning Test-Revised (HVLT-R). EEGs were recorded with grand averaging spectrograms and event-related potentials (ERPs) in rest and cued GO/NOGO task conditions. The results were within the normal range on the MMSE. Full-scale WAIS-III results were in the range of 69–113, with lower scores in verbal understanding than in perceptual organization, and low scores also in indications of working memory and processing speed difficulties. The HVLT-R showed impairment of both immediate and delayed recall. Quantitative EEG showed an increase in low alpha (around 7.5 Hz) activity in the temporofrontal areas, mostly on the left side. There was a deviation in the late (>300 ms) component in response to the NOGO stimuli. A strong correlation (r = 0.78, *p* = 0.01) between the Hopkins Verbal Learning Test (delayed recall) and the amplitude of the NOGO ERP component was observed. The EEG spectra showed deviations from the healthy controls, especially at frontotemporal deviations. Deviations in the ERP component of the NOGO trials were related to delayed recall in the Hopkins Verbal learning test.

## 1. Introduction

Genetic neurological demyelinating disorders affecting both the central and peripheral nervous systems are rare. Hagen et al. (1998) described such a familial disorder with findings indicating demyelination, which differs from other known disorders [1]. This disorder was discovered by Jon Skogholt, who worked as a general practitioner with family members who showed symptoms of the disease [2]. These symptoms are typically gradually developing distal sensory loss, distal atrophy or weakness of extremity muscles, unsteady gait, and—in advanced stages—memory impairment. The onset of the disease appears to be between the third and sixth decades of life. Some family members experienced recurrent episodes of exacerbation, which were initially interpreted as transient cerebral ischemia. Jon Skogholt was responsible for tracing the original pedigree of the family, which was compatible with a maternal gender-linked inheritance. A genetic examination making use of whole-genome sequencing of two patients as compared to two controls was recently carried out, but without identifying the genetic cause. A new approach using the most advanced techniques is in progress, in collaboration with Department of Genetics, Oslo University Hospital. Neuroimaging using magnetic resonance imaging (MRI) of some of the family members revealed white matter lesions, especially in the periventricular areas, and, in all cases, remarkably elevated levels of cerebral fluid protein were observed [1].

Hagen et al. [1] reported the electroencephalographic (EEG) results for four family members. In three of these cases, general slowing was observed, most prominently on the left temporal side of the brain, whereas the EEG was normal in one of these cases [1].

The present study aims to provide a more detailed picture of quantitative EEG (QEEG) and event-related potentials (ERPs), including neuropsychological and neurological data.

The purpose of the study is to obtain a better understanding of this disorder. Our hypothesis is that the disorder is a brain disease and that we will find both neuropsychological (NP) and QEEG/ERP correlates to the clinical and perhaps subclinical manifestations of the disease.

Based on earlier EEG findings, we hypothesize that we will find results indicating more left- than right-side affected areas of the brain.

Our research questions are as follows: Are there NP and QEEG/ERP results which indicate that this disorder is a brain disorder? What kind of NP and QEEG/ERP will we find?

## 2. Subjects and Methods

### 2.1. Subjects

Nine participants were included in this study: five men and four women. They belong to three generations of the affected family and are all affected by the disease.

None of the participants previously had a formal diagnosis of the disease, but there were subtle clinical signs or symptoms in all of them, such as numbness and tendencies for sensory loss in parts of the body, which were considered related to the disorder. In addition to the symptoms, their cerebrospinal fluid (CSF) was examined. A CSF total protein level above 1 g/L (normal range 0.15–0.45 g/L) was defined as sufficient for the diagnosis. A CSF total protein between 0.45 and 1.0 g/L together with clinical symptoms was also considered to be a good enough indicator of the disease.

Since none of the participants had experienced the full development of the disease, we here present 2 cases that did experience the full development of the disease to illustrate the course of the disease.

Subject 1:

From adolescence onwards, this patient experienced episodes of syncope. At the age of 61 years, this patient had a transient attack of mental confusion, combined with headache and vomiting for approximately one day. Physical examination by a general practitioner revealed dysarthria, reduced balance, and modestly impaired movement control of the upper left limb. One year after this episode, the patient experienced another episode of a more severe attack with transient palsy on the left side of her face, confusion, and loss of balance.

From the age of 63, this patient had permanent dysarthria, slow speech, and prolonged latency when responding to questions. The gait was unsteady, and limb movements were slowed. The patient had brisk knee jerks, but bilateral Babinski signs. The patient also had moderate dementia. At that time, hospital specialists interpreted this as recurrent cerebrovascular ischemic attacks.

Subject 2:

From the age of 30, the patient experienced sensory loss and gradually developed loss of control in both arms. At 38 years of age, the general practitioner (Jon Skogholt) observed a slight atrophy of the small hand muscles and mild spasticity in the lower extremities. Sensory modalities were reduced in the upper extremities and deep sensations were impaired in the lower extremities. Tendon reflexes were absent in both arms. The patient had hyperreflexia of the lower extremities. The patient was unable to work after 54 years of age. At that time, medical examination revealed intention tremor, atrophy of the small hand muscles, and distal atrophy of the legs. The patient had impaired coordination in the upper extremities, reduced motor function in all extremities, and an unsteady gait combined with hyperreflexia in the lower extremities.

The control group included 300 healthy subjects (mean age: 48.5 years; range 28–67 years): 129 males and 171 females. The subjects were recruited from three sources in Russia, Switzerland, and Norway (https://www.hbimed.com/en/hbi-database/) (accessed on 1 August 2024). These were all healthy controls (HCs), which included 50 participants of the same age range as the patient group. Subjects with a history of head injury and with neurological or/and psychiatric conditions were excluded from the HC group. The control subjects were not receiving any medication at the time of testing. The database has the approval of both the European (CE Mark) and US authorities (FDA). The selection criteria of the control group have been described in detail in our previous studies [3,4]. Briefly, the following exclusion criteria were applied: (1) any inability to work or to study at school, (2) head injury; (3) long-lasting cases of unconsciousness, (4) epileptic seizures, (5) dementia; (6) hospitalizations due to mental disorders.

### 2.2. Methods

#### 2.2.1. Cognitive Tests

A trained technician tested the participants.

The Mini Mental State Examination (MMSE) [5]: A mental status examination that is often used to evaluate dementia. A score of 30–24 points is indicative of normal functioning. A score below 24 points is often an indication of dementia.The Wechsler Adult Intelligence Scale–Third Edition (WAIS-III) [6,7]: This scale has a mean score of 100. Fifteen points represent one standard deviation. Full-scale IQ scores and four indices are presented.Hopkins Verbal Learning Test [8]: This is a word recall task. The list consists of 12 words. Here, the immediate (three trials) recall and delayed recall (20 min) of the wordlist are reported (age-corrected T-scores). This test may indicate difficulties with verbal memory and was included because patients showed a greater involvement of the left than the right brain hemisphere in disease development.

#### 2.2.2. Cerebral Visualization

SPECT was performed in two participants using a dual-headed gamma camera (E.cam, SIEMENS). The routine acquisition protocol used was a LEHR collimator with 40 s/step, 32 steps. The data were reconstructed according to standard procedures. The cerebellum was used as a reference region. The results were presented as tomographic reconstructions into transverse, coronal, and sagittal slices of perfusion mapping.

A specialist in nuclear medicine conducted the reconstruction and interpretation.

A special scheme was used for the purpose of interpreting the degree of uptake in the different parts of the brain. The type of impairment was described as focal or diffuse, and the uptake was estimated by comparing it to the cerebellar uptake and graded 0–3 (normal, slight, moderate, and significant decrease). In SPECT studies, patients lay down in a quiet, darkened room before examination (acquisition). The eyes were covered but the ears were unplugged. After 10 min of rest, the radiolabelled compound was administered intravenously (740 MBq 99mTc-HMPAO (hexa-methyl-propylene-amine-oxime; Ceretec, GE Healthcare).

#### 2.2.3. EEG Studies

In the EEG, 19 channels were recorded under three conditions: eyes open (3 min), eyes closed (3 min), and a cued GO/NOGO task using the Mitsar 21-channel EEG system (Mitsar, Ltd., St. Petersburg, Russia). Silver/silver chloride electrodes were placed on the skull according to the standard 10–20 system. The input signals referring to the linked ear reference were amplified (bandpass 0.5–30 Hz) and sampled at a rate of 250 Hz. The ground electrode was placed on the forehead. Impedance was maintained below 5 kΩ. EEG signals were further re-referenced to the weighted average reference montage by Lemos [9]. Independent component analysis (ICA) was applied to correct horizontal and vertical eye movement [10].

In addition, epochs with excessive amplitude of filtered EEG and/or excessive faster and/or slower frequency activity were automatically marked and excluded from further analysis. The exclusion thresholds were set as follows, according to previous papers from our group: (a) 100 µV for non-filtered EEG; (b) 50 µV for slow waves in the 0–1 Hz band; and (c) 35 µV for fast waves filtered in the 20–35 Hz band. Finally, the EEG was manually inspected to verify artifact removal. For EEG data analysis, not less than eight artifact-free EEG epochs were used (around 40 s). Before further processing, the entire array of EEG recordings was filtered at the 2–30 Hz frequency band to minimize the overlearning problem in the ICA algorithm.

The executive functions of the brain were assessed using ERP waves in the cued GO/NOGO task [11]. The cued GO/NOGO task was used for studying brain correlates of cognitive control [12,13]. Thus, it fit perfectly for this study. The task comprised 400 trials. Each trial consisted of a sequential presentation of two stimuli. The stimuli were pictures of animals (a), plants (p), and humans (h). They were presented in random order in the following pairs: aa, ap, pp, and ph. The subject’s task consisted of pressing a button to an aa pair. The probabilities for each pair of categories were equal. The intra-stimulus intervals with the pairs were 1000 ms, intervals between pairs were 3000 ms, and the stimulus duration was 100 ms. The button pressing was registered in a special channel, and the reaction time was computed offline.

Spectrograms (using fast Fourier transformation) were computed for three conditions: eyes open, eyes closed, and the task condition. The spectrograms of each individual were collected and compared with the spectrograms of a group of subjects of the same age obtained from the HBI normative database [14]. In addition to spectrograms, ERPs were computed for the cued GO/NOGO task under four different task conditions and compared with normative ERPs.

The statistical comparisons of the patients’ grand-average EEG spectra and ERPs with the corresponding parameters of the healthy controls were made using WinEEG software written for Mitsar-EEG company (https://mitsar-eeg.com/, accessed on 24 January 2023) [15,16].

EEG spectra and ERPs between the two groups were compared using a cluster-based permutation test implemented in WinEEG software (version 03.10.01) [17]. This procedure solved the problem of multiple comparisons by clustering the data based on temporal and spatial proximity. Basically, the cluster-based analysis procedure was similar to the one implemented in the FieldTrip MATLAB toolbox for M/EEG analysis (freely available at http://fieldtrip.fcdonders.nl/ accessed on 5 August 2024 [18]), but differed in the following ways: (1) for comparing ERP waveforms under different conditions, the Wilcoxon signed-rank nonparametric test was used instead of the dependent sample t-tests as in the FieldTrip MATLAB toolbox; (2) for the cluster-level statistics, a normal approximation for the Wilcoxon signed-rank test and the sum z-score within a cluster instead of the sum of the t-values were used. The reason for using nonparametric statistics was their lower sensitivity to outliers.

The data were collected in the same way for the clinical and the control group. The data from the SPECT and NP test results were blinded to the interpreter of the QEEG/ERP.

The 19-channel EEG was digitally filtered in the 0.53–50 Hz frequency band and sampled at 250 Hz. A digital notch filter (45–55 Hz) was used to remove 50 Hz artifacts.

#### 2.2.4. Ethics

Written informed consent was obtained from the patients. The participants consented to the publishing of their case details and any accompanying images. The study participants were allowed to withdraw at any time. Approval of this study was obtained by the Regional Committee for Medical and Health Research Ethics, Region South-East, Norway, ref. no. 556–04224 plus 2013/1017. This study was conducted in accordance with the Declaration of Helsinki.

## 3. Results

Demographics and MMSE, WAIS-III, and HVLT-R results are presented in Table 1.

None of the participants were considered to have dementia and the WAIS-III was more or less within the normal range of two SDs. However, there were indications of impairment in immediate and delayed recall on the HVLT-R.

### 3.1. SPECT

The SPECT results indicated small, patchy reductions in perfusion, mostly in the parietofrontal area, in both participants.

### 3.2. QEEG and ERPs

From a neurological point of view, no EEG abnormalities such as spike/slow wave complexes, paroxysms of slow waves, etc., have been observed.

The grand-averaged spectrograms computed for the cued GO/NOGO task for the group of patients in comparison with the control group are presented in Figure 1.

An excess of EEG power in the band range from 6.5 to 9.5 Hz is clearly seen. This excess activity was distributed over both temporofrontal areas, with maximum activity on the left side. In the healthy controls, spectrograms have a maximum around 9 Hz and are usually distributed over posterior regions, with a maximum in the right occipital temporal area.

The results of the comparison of relative EEG amplitudes for all conditions (eyes open, eyes closed, and cued GO/NOGO task) are presented in Figure 2 In all conditions, there was a statistically significant (*p* < 0.01) increase in relative EEG amplitude over both the temporal and frontal areas, with maximum excess on the left side.

The ERPs in the cued GO/NOGO task compared with the ERPs in the normative group are presented in Figure 3. The statistical analysis of ERP waves showed no deviations in the early (100–200 ms) components. No deviations from the healthy controls were found in the sensory-related components in response to ignored stimuli. No statistically significant deviations were found in the P3 component elicited by the GO stimuli. The late positive NOGO wave induced by NOGO stimuli showed a statistically significant deviation in the late (more than 300 ms) response (Figure 3, left column). This component, distributed over the central frontal areas and generated by a broad area in the anterior cingulate/medial frontal cortex, is suppressed in members of the family.

We observed a correlation between the delayed recall of the HVLT-R and amplitude of the NOGO component in the cued GO/NOGO task. These values were highly correlated (*p* < 0.01). See Figure 4.

## 4. Discussion

In this study, the affected members of the Norwegian family had clinical symptoms indicating a neurological disease. The cognitive evaluation showed results essentially within the normal range on the WAIS-III panel, although some of the participants had difficulties with processing speed. One participant had a total WAIS-III score of 69, which is a borderline result regarding function within the normal range (i.e., two SDs outside the expected mean). The working memory component of the WAIS-III panel also revealed areas with deficits, particularly within the attention and verbal components of the testing. Participants had more difficulties with verbal understanding than perceptual organization. None of the participants were considered to have dementia. The Hopkins Verbal Learning Test showed results indicating difficulties with both immediate and delayed recall. There was essentially no difference between immediate and delayed recall, which is in contrast with the typical Alzheimer’s picture, indicating that there is no typical impairment related to dementia of the Alzheimer’s type. The impairment ranged from a T-score of 50 to 15, with a mean of approximately 30, indicating severe difficulties in learning the wordlist.

The QEEG findings, together with the disseminated minor disturbances in blood perfusion on SPECT, showed principally similar findings regarding deviation from expectation, although more in the parietofrontal location on SPECT rather than in a temporal–frontal location on QEEG.

The cognitive slowness and memory impairment that characterize the present disease may be related to our findings on QEEG, ERPs, and SPECT. The results of QEEG and ERP analyses confirm, to some degree, the results of Hagen et al. [1], with greater difficulty related to the left side of the brain [1]. However, our analysis is more detailed, showing deviance particularly in the alpha band. We also analysed the ERPs in a cued GO/NOGO task. A deviation in the late (>300 ms) response was observed. This late component of the NOGO ERP indicates a deviation from healthy controls over the central–frontal areas of the cortex, associated with a possible lack of suppression from the anterior cingulate/medial frontal cortex. There was a strong correlation between delayed recall in the HVLT-R and the NOGO ERP component at Cz, indicating difficulties with the attentional component of the cued GO/NOGO task and recall. With such a strong correlation, the results indicate attentional difficulties in learning a word list. There was no gender-associated effects in the QEEG or the neuropsychological data.

Similarities in the signs and symptoms within the studied subjects were indicative of an inheritable disease in the present family, seemingly involving peripheral and central demyelination [1,19,20]. Notably, the presentation and heritage of the disease differ from those of multiple sclerosis [1,19]. Furthermore, cerebral autosomal dominant arteriopathy with subcortical infarcts and leukoencephalopathy (CADASIL) was previously excluded [1]. The present findings, together with data from previous studies [1,2], indicate that it is difficult or impossible to classify this disease entity within any defined genetic disorder. Thus, metachromatic leukodystrophy could be ruled out early because of the absence of metachromatic material in sural nerve biopsies [1]. Furthermore, blood plasma samples have demonstrated normal levels of fatty acids, which rule out adrenoleukodystrophy and Krabbe’s disease. The apparent occurrence of minor strokes in affected family members, together with cognitive impairment, is compatible with small-vessel disease, although the rare disease CADASIL has previously been excluded [21]. The central neurological signs depicted are essentially compatible with the signs characterizing Binswanger’s disease, often referred to as subcortical leukoencephalopathy, which is a form of small-vessel/vascular dementia caused by patchy damage to the white brain matter [22]. However, this disease is usually characterized by a more pronounced loss of intellectual function than that revealed in the present family members [23]. It usually presents at an age between 50 and 65 years, with attacks of stroke being characteristic symptoms [24]. Moreover, demyelination in the peripheral nerves in the present family members is peculiar and has not been reported to be a common sign in cerebrovascular diseases. Further research on the present disease is important, as it may increase our knowledge of the development of white matter lesions. It is noteworthy that affected family members have higher levels of copper, iron, and proteins in the cerebrospinal fluid, as compared with the levels in patients or healthy individuals [2]. Apparent shrinkage or underdevelopment of the plica choroidea has recently been reported in patients with this rare disease [19]. Deficient functioning of the blood–brain barrier with a redistribution of proteins, iron, and copper into the central and peripheral nervous tissues may lead to increased oxidative stress, encephalopathy, and, presumably, peripheral demyelination [25,26,27].

In conclusion, the present Norwegian family apparently suffers from a hitherto unrecognized inherited disease involving the CNS, with signs of demyelination and indication changes in brain activity in the frontotemporal areas. The neuropsychological and QEEG/ERP data fit well together; both indicate more left- than right-hemisphere difficulties. The peripheral nervous system is also affected. A mitochondrial-linked inheritance has been suspected, since the pedigree developed by Skogholt shows the transfer of the disease via mothers to the subsequent generation [1]. Genetic studies that focus on this aspect are currently underway. A clear limitation to our study is the small sample size, with only nine participants. This is due to the rarity of the disease, which limits the generalizability of the findings. Also, this study has a cross-sectional design. Future studies should aim to include larger samples, possibly through multicentre collaborations. Tracking the progression of the disease in individual cases over time would provide deeper insights into the natural progression of the disease. A recently published study on this disease disclosed a surprisingly thin plica choroidea barrier, in addition to localized white matter hyperintensities as identified by MRI techniques [18]. These findings will be followed up by more advanced diagnostics, including PET investigations and proteomics of the CSF, both of these techniques being carried out in collaboration with specialists at Oslo University Hospital.

## Figures and Tables

**Figure 1 brainsci-14-00905-f001:**
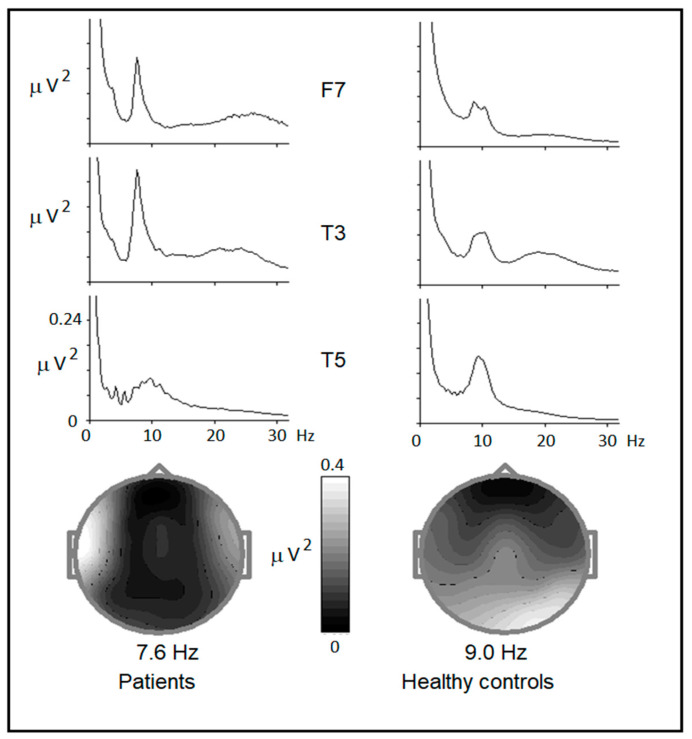
EEG power vs. frequency plots for three electrodes (F7, T3, and T5) for the patient group (**left**) and healthy controls (**right**) of the corresponding age selected from the HBI database (N = 50) with the maps taken at 7.6 and 9.0 Hz, respectively. EEG was recorded during 20 min of the cued GO/NOGO task. Note the asymmetrical lower alpha power at the left frontal–temporal areas of the patient group. *Y*-axis—EEG power in µV^2^. *X*-axis—frequency in Hz from 0 to 30 Hz.

**Figure 2 brainsci-14-00905-f002:**
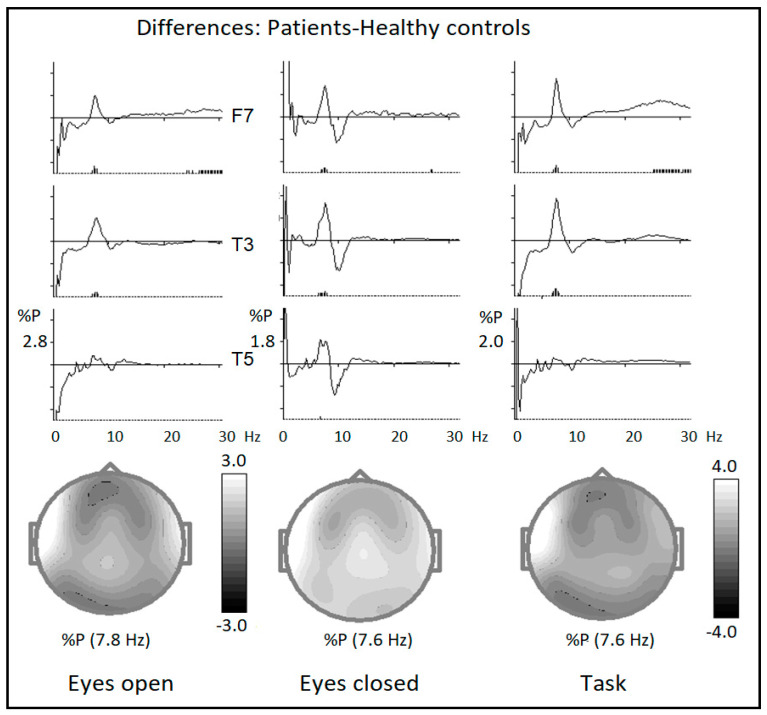
The difference waves (patient group–healthy group) of EEG spectra for three conditions: (**left column**)—eyes open, (**middle column**)—eyes closed, and (**right column**)—cued GO/NOGO task. *Y*-axis—relative EEG power in %. *X*-axis—frequency in Hz from 0 to 30 Hz. Vertical bars below the curves indicate the confidence level of statistical significance of the difference (small bars—*p* < 0.05, larger bars—*p* < 0.01).

**Figure 3 brainsci-14-00905-f003:**
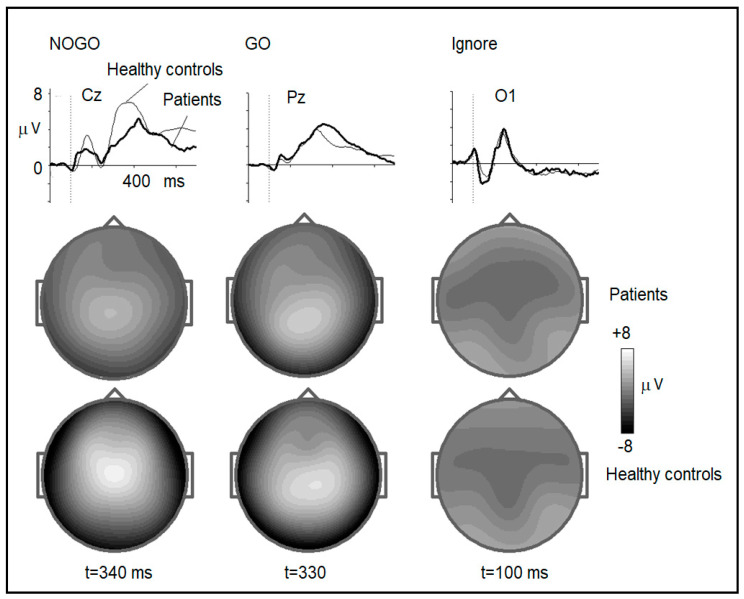
(**Top**): ERPs for NOGO trials (**left**), GO trials (**middle**), and Ignore trials (**right**) averaged over patients (thick lines) and healthy controls (thin lines) for Cz, Pz, and O1 electrodes, respectively. (**Bottom**): Maps of ERPS in NOGO trials (at 340 ms), GO trials (at 330 ms), and Ignore trials (at 100 ms) for the patient and healthy groups. On the plots: Y- axis—averaged potential in µV. X-axis—time after stimulus onset, vertical dotted line—stimulus offset.

**Figure 4 brainsci-14-00905-f004:**
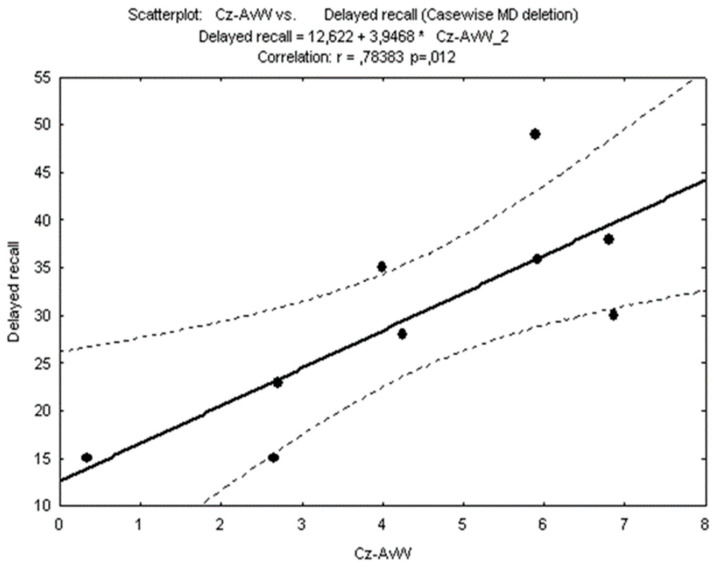
Amplitude of NOGO ERP component at Cz for each patient against T-scores for delayed recall in Hopkins Verbal Learning test-revised. Each patient represented with a black dot.

**Table 1 brainsci-14-00905-t001:** The demographics and Mini Mental State Examination (MMSE), Wechsler Adult Intelligence Scale-III (WAIS-III), and Hopkins Verbal learning test (HVLT-R) results of the study group are presented as mean scores, minimum, maximum, and standard deviations.

Demographics of the Study Group		
Age	46.3	28–67 (12.25)
Education	10.4	7–21 (4.12)
Mental status examination, Intelligence scale results and Memory testing		
Mini Mental State Examination		
MMSE, raw scores	27.7	26–30 (1.64)
Wechsler Adult Intelligence Scale-III		
WAIS-III, Full scale score	90.0	69–113 (14.96)
WAIS-III, Verbal Understanding	89.2	75–122 (14.51)
WAIS-III, Perceptual Organization	100.9	70–135 (17.98)
WAIS-III, Working Memory	87.3	67–108 (13.78)
WAIS-III, Processing Speed	84.7	65–106 (11.08)
Hopkins Verbal Learning test-Revised, T-scores		
Summary Immediate recall	29.8	15–50 (11.6)
Delayed recall	29.6	15–50 (11.5)

## Data Availability

The datasets presented in this article are not readily available because of patient confidentiality. Requests to access the datasets should be directed to Knut Hestad or Jan O. Aaseth.

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
