# Peer review of "Neuropsychological Characteristics and Quantitative Electroencephalography in Skogholt’s Disease—A Rare Neurodegenerative Disease in a Norwegian Family"

_brainsci, 2024, doi:10.3390/brainsci14090905_

Round 1
Reviewer 1 Report
Comments and Suggestions for Authors
Introduction:
- It is necessary to discuss the wider context to justify the study's hypotheses (e.g., neuropsychological research and using QEEG in neurodegenerative diseases with demyelination).
- The purpose and motivation of the study should be expanded (hypotheses, research questions).
Materials and Methods:
- Subjects - based on the pedigree, I conclude that not all 9 subjects had central and peripheral signs. Providing exact numbers and using this breakdown to compare the results may be worthwhile.
- Lines 56 - 60 should be in a figure legend.
- Please describe in detail the selection criteria of the control group in this study.
- Whether similar data acquisition was maintained in all the EEG studies performed (subjects with the disease and a healthy control group)? Could the differences have affected the results, and how?
- Did the SPECT study conducted for the two subjects impact the study's conclusions? These parts of the Method and Results need to be clarified.
- "Spectrogram" is the term used for the time-frequency representation of the signal.
Results:
- Table 1 needs to be better organized - it may not be the best form to show these data, especially for 9 participants. Please also extend the caption.
- Figures - "normals" and "normal group" are improper for scientific writing.
- Figures - Why are there 3 examples of FFT above topomaps?
- The VCPT is mentioned in Figures 2 and 3 and nowhere else in the manuscript. Please explain.
Discussion:
- Please split the first paragraph into smaller ones connected with a single observation/conclusion from the study.
Conclusion:
- There is no conclusion about the neuropsychological characteristics.
- Precise that in "changes in the frontotemporal areas," you mean the brain activity.
Minor:
- repetitions: lines 50-53 - This information is already mentioned in the introduction
- remove the abbreviation (QEEG) from the title
- many formatting errors
Author Response
Review 1
Comments and Suggestions for Authors
Introduction:
- It is necessary to discuss the wider context to justify the study's hypotheses (e.g., neuropsychological
research and using QEEG in neurodegenerative diseases with demyelination).
Our reply: We have added many sentences to the wider context etc. In introduction: Jon Skogholt was
responsible for drawing the original pedigree of the family (Figure 1), which was compatible with a
maternal gender-linked inheritance. A genetic examination making use of a whole genome sequencing
of two patients as compared to two controls was recently carried out, but without identifying the
genetic cause. A new ap-proach using the most advanced techniques is in progress, in collaboration
with Department of Genetics, Oslo University Hospital.
Further down:
The purpose of the study is to get a better understanding of this disorder. Our hypothesis is that the
disorder is a brain disease that we will find both neuropsychological as well as QEEG/ERP correlates
to the clinical and perhaps subclinical manifestations of the disease.
And in Subject and method section: Two typically affected cases that have undergone the full
development of the disease can illustrate the course of the disease:
Further down: The executive functions of the brain were assessed using ERPs waves in the cued
GO/NOGO task [11]. The cued GO/NOGO task was used for studying brain correlates of cognitive
control [12, 13]. Thus, it fit perfectly for this study.
- The purpose and motivation of the study should be expanded (hypotheses, research questions).
Our reply. We have expanded on this: The purpose of the study is to get a better understanding of
this disorder. Our hypothesis is that the disorder is a brain disease that we will find both
neuropsychological as well as QEEG/ERP correlates to the clinical and perhaps subclinical
manifestations of the disease.
Based on earlier EEG findings we hypothesize that we will find results indicating more left than right
side affections of the brain.
Our research questions are: Are there NP and QEEG/ERP results which indicate that this disorder is a
brain disorder? What kind of NP and QEEG/ERP will we find?
Materials and Methods:
- Subjects - based on the pedigree, I conclude that not all 9 subjects had central and peripheral signs.
Providing exact numbers and using this breakdown to compare the results may be worthwhile.
Our reply, we find it difficult to be to exact due to confidentiality, but we have been clearer on that the
two presented cases is not in the study but are there to show the progression of the disease. In
addition we have written: These two participants were not part of the study. For the study we recruited
9 participants from the affected family. None of the subjects in the study had a formal diagnosis of the
disease but there was subtle clinical sign or symptoms as numbness in part of the body, which could
be related to the disorder in all of them.
- Lines 56 - 60 should be in a figure legend.
Our reply: We have followed up.
- Please describe in detail the selection criteria of the control group in this study.
Our reply in Subject and method section: The control group included 300 healthy subjects (mean age
48.5 years; range 28-67 years): 129 males and 171 females. The subjects were recruited from three
sources in Russia, Switzerland, and Norway (https://www.hbimed.com/en/hbi-database/). These were
all healthy controls (HC), which included 50 participants of the same age range as the patient group.
The healthy control subjects were from Chur, Switzerland (HBI med, FDA approved,
https://www.hbimed.com/en/hbi-database/). Subjects with history of head injury, with neurological
or/and psychiatric conditions were excluded from the HBI med group. The control subjects were not
receiving any medication at the time of testing. The database has the approval of both the European
(CE Mark) and the US authorities (FDA). The selection criteria of the control group have been
described in detail in our previous studies[3, 4]. Briefly, the following exclusion criteria were ap-plied:
1) any inability to work or to study at school, 2) head injury; 3) long lasting cases of unconsciousness,
4) epileptic seizures, 5) dementia; 6) hospitalizations due to mental disorders.
- Whether similar data acquisition was maintained in all the EEG studies performed (subjects with the
disease and a healthy control group)? Could the differences have affected the results, and how?
Our reply: The data were collected in the same way in the clinical and the control group
- Did the SPECT study conducted for the two subjects impact the study's conclusions? These parts of
the Method and Results need to be clarified.
Our reply with the following text: The data from the SPECT and NP test results was blinded to the to
the interpreter of the QEEG/ERP.
- "Spectrogram" is the term used for the time-frequency representation of the signal.
See our description of both QEEG methods and figures.
Results:
- Table 1 needs to be better organized - it may not be the best form to show these data, especially for
9 participants. Please also extend the caption.
Our reply: Hope it is better organized now
- Figures - "normals" and "normal group" are improper for scientific writing.
We have changed the writing. Will use healthy control group.
- Figures - Why are there 3 examples of FFT above topomaps?
Our reply: Much of the QEEG part is rewritten.
- The VCPT is mentioned in Figures 2 and 3 and nowhere else in the manuscript. Please explain.
Our reply: We have changed this. See capitation of the figures.
Discussion:
- Please split the first paragraph into smaller ones connected with a single observation/conclusion from
the study.
Our reply: This is done.
Conclusion:
- There is no conclusion about the neuropsychological characteristics.
Our reply: The neuropsychological and QEEG/ERP data fit well together; both indicate more left than
right hemisphere difficulties.
- Precise that in "changes in the frontotemporal areas," you mean the brain activity.
Our reply: Yes, and we tried to be more precise here.
Minor:
- repetitions: lines 50-53 - This information is already mentioned in the introduction
- remove the abbreviation (QEEG) from the title
- many formatting errors
Our reply: Hope it is better now.

Reviewer 2 Report
Comments and Suggestions for Authors
Dear Author,
The manuscript entitled “Neuropsychological Characteristics and Quantitative Electroencephalography (QEEG) in Skogholt’s Disease - A Rare Neurodegenerative Disease in a Norwegian Family” detailed investigation into Skogholt’s disease, a rare neurodegenerative disorder, through neuropsychological assessments, SPECT imaging, and QEEG/ERP analysis. The study aims to elucidate the cognitive and neurophysiological characteristics of the disease in a Norwegian family. The study addresses a rare neurodegenerative disorder, Skogholt’s disease, which has not been extensively documented. This adds significant value to the field of neurology and neurogenetics by providing detailed insights into its neuropsychological and neurophysiological characteristics. The authors employed a robust and comprehensive methodology, including neuropsychological assessments (MMSE, WAIS-III, HVLT-R), SPECT imaging, and advanced EEG/ERP analysis. The inclusion of multiple assessment tools strengthens the validity of the findings. The discussion section effectively contextualizes the findings within the broader literature, highlighting the unique aspects of Skogholt’s disease and suggesting avenues for further research. The potential genetic basis and the hypothesis regarding mitochondrial-linked inheritance are particularly intriguing. Overall, this manuscript offers a valuable contribution to the understanding of Skogholt’s disease, but addressing the noted limitations and expanding certain sections would significantly enhance its impact.
- The study's sample size is relatively small, with only nine participants. While this is understandable given the rarity of the disease, it limits the generalizability of the findings. Future studies should aim to include larger samples, possibly through multi-center collaborations.
- The control group, although well-defined, could be better matched to the patient group in terms of demographics, particularly age and education level. This would help to control for potential confounding variables.
- While the study mentions ongoing genetic research, the current manuscript does not provide detailed genetic analysis results. Including preliminary genetic findings would enhance the understanding of the disease's hereditary nature.
- The SPECT imaging results are briefly mentioned but not thoroughly discussed or visualized in the manuscript. Providing more detailed descriptions and visual representations of the SPECT findings would strengthen the manuscript.
- The case reports, while informative, could benefit from more detailed longitudinal data. Tracking the progression of the disease in individual cases over time would provide deeper insights into its natural history.
- Some technical aspects of the QEEG and ERP methodologies could be elaborated upon, particularly regarding the preprocessing steps and the rationale for certain parameter choices. This would improve the clarity and reproducibility of the neurophysiological analyses.
- The discussion section could benefit from a more thorough consideration of the study's limitations, including the small sample size, potential selection bias, and the challenges in diagnosing rare diseases. Acknowledging these limitations would provide a more balanced view of the study's contributions.
- While the manuscript suggests further research is needed, it could be more specific about the future directions. For instance, outlining specific hypotheses or potential interventions based on the current findings would provide a clearer roadmap for subsequent studies.
Author Response
Review 2:
Dear Author,
The manuscript entitled “Neuropsychological Characteristics and Quantitative Electroencephalography
(QEEG) in Skogholt’s Disease - A Rare Neurodegenerative Disease in a Norwegian Family” detailed
investigation into Skogholt’s disease, a rare neurodegenerative disorder, through neuropsychological
assessments, SPECT imaging, and QEEG/ERP analysis. The study aims to elucidate the cognitive and
neurophysiological characteristics of the disease in a Norwegian family. The study addresses a rare
neurodegenerative disorder, Skogholt’s disease, which has not been extensively documented. This
adds significant value to the field of neurology and neurogenetics by providing detailed insights into its
neuropsychological and neurophysiological characteristics. The authors employed a robust and
comprehensive methodology, including neuropsychological assessments (MMSE, WAIS-III, HVLT-R),
SPECT imaging, and advanced EEG/ERP analysis. The inclusion of multiple assessment tools
strengthens the validity of the findings. The discussion section effectively contextualizes the findings
within the broader literature, highlighting the unique aspects of Skogholt’s disease and suggesting
avenues for further research. The potential genetic basis and the hypothesis regarding mitochondriallinked inheritance are particularly intriguing. Overall, this manuscript offers a valuable contribution to the
understanding of Skogholt’s disease, but addressing the noted limitations and expanding certain
sections would significantly enhance its impact.
Our reply in the Discussion part, Conclusion: In conclusion, the present Norwegian family apparently
suffers from a hitherto unrecognized inherited disease involving the CNS, with signs of demyelination
and indication changes of brain activity in the frontotemporal areas. The neuropsychological and
QEEG/ERP data fit well together; both indicate more left than right hemisphere difficulties. The
peripheral nervous system was also affected. A mitochondrial-linked inheritance has been suspected,
since the pedigree (figure 1) shows the transfer of the disease via mothers to the subsequent generation.
Genetic studies that focus on this aspect are currently underway. A clear limitation to our study is the
small sample size, with only nine participants. This is due to the rarity of the disease, which limits the
generalizability of the findings. Also, and it was a cross-sectional design. Future studies should aim to
include larger samples, possibly through multicentre collaborations. Tracking the progression of the
disease in individual cases over time would provide deeper insights into the natural progression of the
disease. A recently published study on this disease disclosed a surprisingly thin plica choroidea barrier,
in addition to localized white matter hyperintensities as identified by MRI-techniques [18]. These findings will be followed up by more advanced diagnostics including PET investigations and proteinomics
of CSF, both these techniques being carried out in collaboration with specialists at Oslo University
Hospital.
- The study's sample size is relatively small, with only nine participants. While this is understandable
given the rarity of the disease, it limits the generalizability of the findings. Future studies should aim to
include larger samples, possibly through multi-center collaborations.
Our reply: See above, we have included the suggestion from the reviewer.
- The control group, although well-defined, could be better matched to the patient group in terms of
demographics, particularly age and education level. This would help to control for potential confounding
variables.
Our reply: We have expanded on the control group: The control group included 300 healthy subjects
(mean age 48.5 years; range 28-67 years): 129 males and 171 females. The subjects were recruited
from three sources in Russia, Switzerland, and Norway (https://www.hbimed.com/en/hbi-database/).
These were all healthy controls (HC), which included 50 participants of the same age range as the
patient group. The healthy control subjects were from Chur, Switzerland (HBI med, FDA approved,
https://www.hbimed.com/en/hbi-database/). Subjects with history of head injury, with neurological
or/and psychiatric conditions were excluded from the HBI med group. The control subjects were not
receiving any medication at the time of testing. The database has the approval of both the European
(CE Mark) and the US authorities (FDA). The selection criteria of the control group have been described
in detail in our previous studies[3, 4]. Briefly, the following exclusion criteria were ap-plied: 1) any
inability to work or to study at school, 2) head injury; 3) long lasting cases of unconsciousness, 4)
epileptic seizures, 5) dementia; 6) hospitalizations due to mental disorders.
We have specified that 50 of the controls have the same age range as the 9 participants.
- While the study mentions ongoing genetic research, the current manuscript does not provide detailed
genetic analysis results. Including preliminary genetic findings would enhance the understanding of the
disease's hereditary nature.
Our reply, in the introduction: Jon Skogholt was responsible for drawing the original pedigree of the
family (Figure 1), which was compatible with a maternal gender-linked inheritance. A genetic
examination making use of a whole genome sequencing of two patients as compared to two controls
was recently carried out, but without identifying the genetic cause. A new approach using the most
advanced techniques is in progress, in collaboration with Department of Genetics, Oslo University
Hospital.
And in the conclusion: Genetic studies that focus on this aspect are currently underway. A clear limitation
to our study is the small sample size, with only nine participants. This is due to the rarity of the disease,
which limits the generalizability of the findings. Also, and it was a cross-sectional design. Future studies
should aim to include larger samples, possibly through multicentre collaborations. Tracking the
progression of the disease in individual cases over time would provide deeper insights into the natural
progression of the disease.
- The SPECT imaging results are briefly mentioned but not thoroughly discussed or visualized in the
manuscript. Providing more detailed descriptions and visual representations of the SPECT findings
would strengthen the manuscript.
Our reply: We could have done what the reviewer suggests, but it is only in two participants and the
deviation from the norm is very small. We have written: The SPECT results indicated patchy small
reductions in perfusion, mostly in the parieto-frontal area, in both participants.
- The case reports, while informative, could benefit from more detailed longitudinal data. Tracking the
progression of the disease in individual cases over time would provide deeper insights into its natural
history.
Our reply: We agree.
- Some technical aspects of the QEEG and ERP methodologies could be elaborated upon, particularly
regarding the preprocessing steps and the rationale for certain parameter choices. This would improve
the clarity and reproducibility of the neurophysiological analyses.
Our reply. We have expanded the QEEG and ERP methodologies: The executive functions of the brain
were assessed using ERPs waves in the cued GO/NOGO task [11]. The cued GO/NOGO task was used
for studying brain correlates of cognitive control [12, 13]. Thus, it fit perfectly for this study. The task
comprised 400 trials. Each trial consisted of a sequential presentation of two stimuli. The stimuli were
pictures of animals (a), plants (p), and humans (h). They were presented in random order in the following
pairs: aa, ap, pp, and ph, with a subject’s task of pressing a button to an aa pair. The probabilities for
each pair of categories were equal. The in-tra-stimulus intervals with the pairs were 1000 milli secund,
intervals between pairs were 3000 milli secund, and the stimulus duration was 100 milli secund. The
button pressing was registered in a special channel, and the reaction time was computed offline.
Spectrograms (using fast Fourier transformation) were computed for three condi-tions: eyes open, eyes
closed, and task condition. The spectrograms of each individual were collected and compared with the
spectrograms of a group of subjects of the same age obtained from the HBI normative database [14].
In addition to spectrograms, ERPs were computed for the cued GO/NOGO task under four different task
conditions and compared with normative ERPs.
The statistical comparisons of the patients’ grand average EEG spectra and ERPs with the
corresponding parameters of the healthy controls were made by WinEEG software written for MitsarEEG company (https://mitsar-eeg.com/ 24.01.2023) [15, 16].
EEG spectra and ERPs between the two groups were compared using a cluster-based permutation test
implemented in WinEEG software[17]. This procedure solved the problem of multiple comparisons by
clustering the data based on temporal and spatial proximity. Basically, the cluster-based analysis
procedure was similar to the one im-plemented in the FieldTrip MATLAB toolbox for M/EEG analysis
(freely available at http://fieldtrip.fcdonders.nl/ [18], but differed in the following: 1) for comparing ERP
waveforms under different conditions the Wilcoxon signed-rank nonparametric test was used instead of
the dependent sample t-tests as in FieldTrip MATLAB toolbox; 2) for the cluster-level statistics a normal
approximation for the Wilcoxon signed rank test and the sum z-score within a cluster instead of the sum
of the t-values were used. The reason for using nonparametric statistics was their less sensitivity to
outliers.
- The discussion section could benefit from a more thorough consideration of the study's limitations,
including the small sample size, potential selection bias, and the challenges in diagnosing rare diseases.
Acknowledging these limitations would provide a more balanced view of the study's contributions.
Our reply, see our conclusion: A clear limitation to our study is the small sample size, with only nine
participants. This is due to the rarity of the disease, which limits the generalizability of the findings. Also,
and it was a cross-sectional design. Future studies should aim to include larger samples, possibly
through multicentre collaborations. Tracking the progression of the disease in individual cases over time
would provide deeper insights into the natural progression of the disease.
- While the manuscript suggests further research is needed, it could be more specific about the future
directions. For instance, outlining specific hypotheses or potential interventions based on the current
findings would provide a clearer roadmap for subsequent studies.
Our reply, also in conclusion: Future studies should aim to include larger samples, possibly through
multicentre collaborations. Tracking the progression of the disease in individual cases over time would
provide deeper insights into the natural progression of the disease. A recently pub-lished study on
this disease disclosed a surprisingly thin plica choroidea barrier, in ad-dition to localized white matter
hyperintensities as identified by MRI-techniques [18]. These findings will be followed up by more
advanced diagnostics including PET investigations and proteinomics of CSF, both these techniques
being carried out in collaboration with specialists at Oslo University Hospital.

Reviewer 3 Report
Comments and Suggestions for Authors
Dear Authors,
several points of the manuscript need to be better clarified.
Do the nine participants studied correspond to the patients in the pedigree (where only 7 patients are alive)? Do the two subjects described (lines 70-93) belong to the population studied, and do they correspond to the patients mentioned on line 188?
The figures and captions are not optimally informative. In figures 2 and 3 there is no indication of the units of measurement on the x-axis. Figure 1: “Anonymous” (?). Line 191: “… in figure 1” (?).
EEG. Did the visual analysis of the EEGs subjected to quantitative evaluation highlight any anomalies?
Evoked potentials. The protocol used for event-RPs must be better detailed, both in terms of the stimulation paradigm and the components studied. Which bandpass was used?
Skogholt's disease is a demyelinating disease. I therefore hypothesize that the patients underwent study of stimulus related potentials (somatosensory and visual). Taking into account the stimulation paradigm of event-RPs, marked alterations of visual evoked potentials can influence the characteristics of subsequent cognitive components. It is therefore appropriate to accompany the manuscript with a description, albeit brief, of the characteristics of the stimulus-related evoked potentials in the patients studied.
Best regards.
Author Response
Review 3:
Do the nine participants studied correspond to the patients in the pedigree (where only 7
patients are alive)? Do the two subjects described (lines 70-93) belong to the population
studied, and do they correspond to the patients mentioned on line 188?
Our reply: We have tried to make it clearer that the two participants described are not part of the
nine participants included in the study. Two of the participants included belongs to the next
generation who are not part of the of the pedigree. See our text in the description of the
participants: These two participants were not part of the study. For the study we recruited 9
participants for the affected family. None of the subjects in the study had a formal diagnosis of
the disease but there was subtle clinical sign or symptoms as numbness in part of the body,
which could be related to the disorder in all of them.
The figures and captions are not optimally informative. In figures 2 and 3 there is no indication of
the units of measurement on the x-axis. Figure 1: “Anonymous” (?). Line 191: “… in figure 1” (?).
Our reply: This is now changed. See the capitation to the figures.
EEG. Did the visual analysis of the EEGs subjected to quantitative evaluation highlight any
anomalies?
Our reply. No, they did not. See our text in the beginning of the Result section, under the
heading: QEEG and ERP’s. From neurological point of view no EEG abnormalities such as
spike/slow wave complexes, paroxysms of slow waves, etc. have been observed.
Evoked potentials. The protocol used for event-RPs must be better detailed, both in terms of the
stimulation paradigm and the components studied. Which bandpass was used?
Skogholt's disease is a demyelinating disease. I therefore hypothesize that the patients
underwent study of stimulus related potentials (somatosensory and visual). Taking into account
the stimulation paradigm of event-RPs, marked alterations of visual evoked potentials can
influence the characteristics of subsequent cognitive components. It is therefore appropriate to
accompany the manuscript with a description, albeit brief, of the characteristics of the
stimulus-related evoked potentials in the patients studied.
Our reply for the last two paragraphs, in the description of the ERPs: The executive functions of
the brain were assessed using ERPs waves in the cued GO/NOGO task [11]. The cued GO/NOGO
task was used for studying brain correlates of cognitive control [12, 13]. Thus, it fit perfectly for
this study. The task comprised 400 trials. Each trial consisted of a sequential presentation of two
stimuli. The stimuli were pictures of animals (a), plants (p), and humans (h). They were
presented in random order in the following pairs: aa, ap, pp, and ph, with a subject’s task of
pressing a button to an aa pair. The probabilities for each pair of categories were equal. The intra-stimulus intervals with the pairs were 1000 milli secund, intervals between pairs were 3000
milli secund, and the stimulus duration was 100 milli secund. The button pressing was
registered in a special channel, and the reaction time was computed offline.
Spectrograms (using fast Fourier transformation) were computed for three conditions: eyes
open, eyes closed, and task condition. The spectrograms of each individual were collected and
compared with the spectrograms of a group of subjects of the same age obtained from the HBI
normative database [14]. In addition to spectrograms, ERPs were computed for the cued
GO/NOGO task under four different task conditions and compared with normative ERPs.

Reviewer 4 Report
Comments and Suggestions for Authors
brainsci-3072318: “Neuropsychological characteristics and Quantitative electroencephalography (QEEG) in Skogholt’s disease - a rare neurodegenerative disease in a Norwegian family”.
The authors describe neuropsychological and electroencephalographic characteristics of Norwegian family members with a rare demyelinating disease identified in their three generations.
This study adds a new knowledge about differentiation of different neurodegenerative diseases based on a specificity of cognitive impairments, in particular, dementia.
The material, presented in this paper, is unique and needs further studies comparing other neurodegenerative pathologies.
To analyse a specificity of the Skogholt’s disease mechanisms further studies should include their detailed comparing with those of more “familiar” pathologies as Alzheimer disease, multiple and amyotrophic lateral sclerosis.
The main question posed in the study was addressed; the conclusions are partly consistent with the evidence and arguments presented
Minor remarks/recommendations:
1) in line 3, “(QEEG)” should be removed;
2) in lines 21-23, “A strong correlation (r = .78, p =. 01) between the Hopkins Verbal Learning Test (delayed recall) and the amplitude of the NOGO ERP component was observed.” looks better;
3) in line 39, “...gender-linked inheritance.” looks better;
4) in line 43, “(EEG)” should be added;
5) in line 53, a reference should be added;
6) in line 59 and everywhere in the text, “...with coauthors...” should be used;
7) in line 85, “(JS)” should be either replaced by (“GP”) or removed;
8) in line 118, “...slight, moderate and....”;
9) in lines 125 and 126, the equipment reference needs more information;
10) in lines 130 and 131, “...by Lemos [9]. Independent component analysis (ICA) was...”;
11) in line 140, “...in the ICA algorithm...”;
12) in Figure 2, each plate should be denoted by a letter and described in the legend;
13) in Figure 2, horizontal axes for EEG spectrograms need detailed measurement units;
14) in lines 202 and 203, the sentence of “These results were consistent.” should be either removed or rewritten;
15) in Figure 3, each plate should be denoted by a letter and described in the legend;
16) in Figure 3, the axes should be entitled, and units should be added;
17) in Figure 4, each plate should be denoted by a letter and described in the legend;
18) in Figure 4, the fonts should be bigger;
19) in line 258, “There were no gender-associated effects in the QEEG or the neuropsychological data.” looks better;
20) in lines 260-262, “Similarities in the signs and symptoms within the studied subjects were indicative of an inheritable disease in the present family seemingly involving peripheral and central demyelination [1, 17, 18].” looks better;
21) in line 284, “...in patients...” needs details;
22) the References list should be formatted correctly.
Comments on the Quality of English Language
Minor editing of English language required
Author Response
brainsci-3072318: “Neuropsychological characteristics and Quantitative electroencephalography (QEEG) in
Skogholt’s disease - a rare neurodegenerative disease in a Norwegian family”.
The authors describe neuropsychological and electroencephalographic characteristics of Norwegian family
members with a rare demyelinating disease identified in their three generations.
This study adds a new knowledge about differentiation of different neurodegenerative diseases based on a
specificity of cognitive impairments, in particular, dementia.
The material, presented in this paper, is unique and needs further studies comparing other
neurodegenerative pathologies.
To analyse a specificity of the Skogholt’s disease mechanisms further studies should include their detailed
comparing with those of more “familiar” pathologies as Alzheimer disease, multiple and amyotrophic lateral
sclerosis.
The main question posed in the study was addressed; the conclusions are partly consistent with the evidence
and arguments presented
Minor remarks/recommendations:
1) in line 3, “(QEEG)” should be removed;
Our reply: it has been removed.
2) in lines 21-23, “A strong correlation (r = .78, p =. 01) between the Hopkins Verbal Learning Test
(delayed recall) and the amplitude of the NOGO ERP component was observed.” looks better;
Our reply: We have followed up.
3) in line 39, “...gender-linked inheritance.” looks better;
Our reply: We have followed up.
4) in line 43, “(EEG)” should be added;
Our reply: We have followed up.
5) in line 53, a reference should be added;
Our reply: See also text to the figure. It should be clear that the reference is Hagen et al.
6) in line 59 and everywhere in the text, “...with coauthors...” should be used;
Our reply: We use the term et al. Should that be changed?
7) in line 85, “(JS)” should be either replaced by (“GP”) or removed;
Our reply: We have followed up.
We have made it clear that JS is Jon Skogholt, We have spelled it out to get rid of misunderstandings.
8) in line 118, “...slight, moderate and....”;
Our reply: We have followed up.
9) in lines 125 and 126, the equipment reference needs more information;
Our reply: Hope it is good enough now.
10) in lines 130 and 131, “...by Lemos [9]. Independent component analysis (ICA) was...
Our reply: We have followed up.
11) in line 140, “...in the ICA algorithm...”;
Our reply: We have followed up.
12) in Figure 2, each plate should be denoted by a letter and described in the legend;
Our reply: We have followed up.
13) in Figure 2, horizontal axes for EEG spectrograms need detailed measurement units;
14)
Our reply: We have followed up. See capitation for the figures.
15) in lines 202 and 203, the sentence of “These results were consistent.” should be either removed or
rewritten;
Our reply: We have followed up
15) in Figure 3, each plate should be denoted by a letter and described in the legend;
16) in Figure 3, the axes should be entitled, and units should be added;
17) in Figure 4, each plate should be denoted by a letter and described in the legend;
18) in Figure 4, the fonts should be bigger;
Our reply: We have followed up 15-18 especially in the Capitation of the figures.
19) in line 258, “There were no gender-associated effects in the QEEG or the neuropsychological data.” looks
better;
Our reply. We have followed up on page 11. There was no gender-associated effects in the QEEG or
the neuropsychological data.
20) in lines 260-262, “Similarities in the signs and symptoms within the studied subjects were indicative of an
inheritable disease in the present family seemingly involving peripheral and central demyelination [1, 17, 18].”
looks better;
Our reply: We have followed up
21) in line 284, “...in patients...” needs details;
Our reply: We have followed up and included: It is noteworthy that affected family members have higher levels
of copper, iron, and proteins in the cerebrospinal fluid, as compared with the levels in patients or healthy
individuals [2].
22) the References list should be formatted correctly.
Our reply: Hope the references is ok now.
Comments on the Quality of English Language
Minor editing of English language required
Our reply: Hope the language is ok now

Round 2
Reviewer 3 Report
Comments and Suggestions for Authors
unfortunately, the responses provided by the authors to the comments made and the revisions made to the text of the manuscript did not improve the quality of the article. On the contrary.
The diagnosis of Skogholt's disease in the nine subjects studied is hypothetical. It does not appear that any instrumental tests capable of increasing the degree of diagnostic probability have been carried out. (While the study would be much more convincing if carried out in certainly affected subjects). In particular, no answer is provided regarding the fact that the subjects studied were subjected - in addition to ERPs - to recording of stimulus-related evoked potentials, which could contribute to confirming the hypothesized diagnosis. It does not appear that the comments made regarding evoked potentials have been fully taken into consideration (for example, nothing is said about the bandpass used for recording ERPs).
Do some of the subjects studied belong to the pedigree presented? It seems appropriate to present a pedigree that also includes the "next generation". Or, to avoid confusion, eliminate the figure of the incomplete pedigree from the article.
For the reasons now explained, in my opinion, the quality of the manuscript makes its publication inappropriate.
Author Response
The reviewer’s comments:
The diagnosis of Skogholt's disease in the nine subjects studied is
hypothetical. It does not appear that any instrumental tests capable of
increasing the degree of diagnostic probability have been carried out.
(While the study would be much more convincing if carried out in
certainly affected subjects). In particular, no answer is provided
regarding the fact that the subjects studied were subjected - in addition
to ERPs - to recording of stimulus-related evoked potentials, which
could contribute to confirming the hypothesized diagnosis. It does not
appear that the comments made regarding evoked potentials have been
fully taken into consideration (for example, nothing is said about the
bandpass used for recording ERPs).
Do some of the subjects studied belong to the pedigree presented? It
seems appropriate to present a pedigree that also includes the "next
generation". Or, to avoid confusion, eliminate the figure of the
incomplete pedigree from the article.
AUTHORS: Our reply regaring the dignosis of the disese. We were a little reluctant
to present all data for diagnosis together with the pedigree of family,
which make it easy to recognize the particular person. Also, some of the
participants of the study belong to a generation that were not in the
pedigree. Therefore we agree with the reviewer to avoid confusion, by
eliminating the figure of the incomplete pedigree from the article. To
come closer to a diagnosis of the participants we also present CSF results
of the participants presented.
2.1. Subjects
Nine participants were included in the study, 5 men and 4 females.
They are belonging to three generations of the affected family and were
all affected by the disease.
None of the participants had previously a formal diagnosis of the
disease but there were subtle clinical signs or symptoms as numbness and
tendencies to sensory loss in part of the body, which were considered
related to the disorder in all of them. AUTHORS REVISION: They all belonged to the affected
family. In addition to the symptoms, their cerebrospinal fluid (CSF) was
examined. A CSF total-protein level above 1 g/L (normal range 0.15-0.45
g/L) was defined as sufficient for the diagnosis. A CSF total-protein
between 0.45–1.0 g/L together with clinical symptoms was also considered
to be good enough indicator of the disease.
Since none of the participants had full blown development of the
disease, we here present 2 cases that have undergone the full development
of the disease to illustrate the course of the disease:
Subject 1:
From adolescence onwards this subject experienced episode of
syncope. At the age of 61 years, this patient had a transient attack of mental
confusion, combined with headache and vomiting for approximately one
day. Physical examination by a general practitioner revealed dysarthria,
reduced balance, and modestly impaired movement control of the upper
left limb. One year after this episode, the patient experienced another
episode of a more severe attack with transient palsy on the left side of her
face, confusion, and loss of balance.
From the age of 63, this patient had permanent dysarthria, slow
speech, and prolonged latency when responding to questions. The gait
was unsteady and limb movements were retarded. The patient had brisk
knee jerks, but bilateral Babinski signs. The patient also had moderate
dementia. At that time, hospital specialists interpreted his disease as a
recurrent cerebrovascular ischemic attack.
Subject 2:
From the age of 30, the patient experienced sensory loss and
gradually developed loss of control in both arms. At 38 years of age, the
general practitioner (Jon Skogholt) observed slight atrophy of the small
hand muscles and mild spasticity in the lower extremities. Sensory
modalities were reduced in the upper extremities and deep sensations
were impaired in the lower extremities. Tendon reflexes were absent in
both arms. The patient had hyperreflexia of the lower extremities. The
participants were unable to work after 54 years of age. At that time,
medical examination revealed intention tremor, atrophy of the small hand
muscles, and distal atrophy of the legs. She had impaired coordination in
the upper extremities, reduced motor function in all extremities, and an
unsteady gait combined with hyperreflexia in the lower extremities.
AUTHORS Regarding the QEEG and ERP we have filled in:
19-channel EEG was digitally filtered in 0.53-50 Hz frequency band and
sampled at 250 Hz. A digital notch filter (45–55 Hz) was used to remove
50 Hz artifact.